


# Source apportionment of atmospheric ammonia before, during, and after the 2014 APEC summit in Beijing using stable nitrogen isotope signatures

Yunhua Chang[1, 2], Xuejun Liu[3], Congrui Deng[1], Anthony J. Dore[4], and Guoshun Zhuang[1]

[1]Center for Atmospheric Chemistry Study, Department of Environmental Science and Engineering, Fudan University, Shanghai 200433, China

[2]School of Environmental Science and Engineering, Shaanxi University of Science and Technology, Xi'an, Shaanxi 710021, China

[3]College of Resources and Environmental Sciences, China Agricultural University, Beijing 100193, China

[4]Centre for Ecology & Hydrology Edinburgh, Bush Estate, Penicuik, Midlothian EH26 0QB, UK

*Correspondence to*: Guoshun Zhuang (gzhuang@fudan.edu.cn), Congrui Deng (congruideng@fudan.edu.cn) and Yunhua Chang (changy13@fudan.edu.cn)

**Abstract.** Stable nitrogen isotope composition ($\delta^{15}N$) offers new opportunities to address the long-standing and ongoing controversy regarding the origins of ambient ammonia ($NH_3$), a vital precursor of $PM_{2.5}$ inorganic components, in the urban atmosphere. In this study, the $\delta^{15}N$ values of $NH_3$ samples collected from various sources were constrained using a novel and robust chemical method coupled with standard elemental analysis procedures. Independent of the wide variation in mass concentrations (ranging from 33 (vehicle) to over 6000 (human excreta) µg m$^{-3}$), different $NH_3$ sources have generally different $\delta^{15}N$ values (ranging from -52.0 to -9.6‰). Significantly high $\delta^{15}N$ values are seen as a characteristic feature of all vehicle-derived $NH_3$ samples (-14.2±2.8‰), which can be distinguished from other sources emitted at environmental temperature (-29.1±1.7, -37.8±3.6, and -50.0±1.8‰ for livestock, waste, and fertilizer, respectively).

The isotope $\delta^{15}N$ signatures for a range of $NH_3$ emission sources were used to evaluate the contributions of the different sources within measured ambient $NH_3$ in Beijing, using an isotope mixing model (IsoSource). The method was used to quantify the sources of ambient $NH_3$ before, during and after the 2014 APEC summit, when a set of stringent air quality control measures were implemented. Results show that the average $NH_3$ concentrations (the overall contributions of traffic, waste, livestock, and fertilizer) during the three periods were 9.1 (15.1, 31.2, 23.7, and 30.0%), 7.3 (8.8, 24.9, 14.3, and





52.0%), and 12.7 (29.4, 23.6, 31.7, and 15.4%) µg m$^{-3}$, respectively, representing a 20.0% decrease first and then a 74.5% increase in overall NH$_3$ mass concentrations. During (after) the summit, the contributions of traffic, waste, livestock, and fertilizer decreased (increased) by 58.7 (234.2), 0.9 (-5.0), 41.0 (120.8), and -87.6% (-70.5%) when compared with periods before (during) the summit, respectively, signifying that future NH$_3$ control efforts in megacities like Beijing should prioritize traffic sector as well as livestock breeding. The results show that isotope ratio measurements of NH$_3$ to be a valuable tool to quantify the atmospheric sources of NH$_3$ in urban atmospheres.

## 1 Introduction

For more than a century, the Haber-Bosch process has been meeting the Earth's increasing demand for grain and protein through nitrogen fertilizer (Erisman et al., 2008; Sutton et al., 2011). But unintentionally, increasing the application of ammonia (NH$_3$) and its derivatives as fertilizer has altered the composition of the atmosphere (Fowler et al., 2009). Fertilizer application merged with livestock production is the largest contributor of NH$_3$ emissions (Aneja et al., 2008) which cause cascading effects on human health, ecosystems, and climate (Galloway et al., 2003).

Whilst the overwhelming contribution of agricultural activities to the global and regional NH$_3$ budgets is well accepted, a large number of observations show that the ambient levels of NH$_3$ concentration in cities are comparable with, or even higher than those in rural areas (e.g., Cao et al., 2009; Meng et al., 2011; Singh and Kulshrestha, 2014; Zbieranowski and Aherne, 2012). Given that the atmospheric behavior of NH$_3$ is characterized by a short lifetime (1-5 days or less) (Warneck, 1999), low transport height, and relatively high dry deposition velocity (Asman and van Jaarsveld, 1992), high rural NH$_3$ emissions do not generally influence extensive urban areas strongly in the gaseous phase unless reacting with acidic gases locally to form particulate NH$_4^+$ (Flechard et al., 2013). Therefore, other non-agricultural sources must exist in urban areas (Chang, 2014; Sutton et al., 2000). Some authors argue that gasoline-powered vehicles equipped with three-way catalytic converters (TWCs) and diesel-powered vehicles fitted with the selective catalytic reduction system (SCRs) are a major contributor of non-agricultural NH$_3$ (Burgard et al., 2006; Liu et al., 2014c; Perrino et al., 2002). In the UK, for example, it is estimated that 15% of the national NH$_3$ emissions originate from non-agricultural activities (Sutton et al., 2000), and 5-6% of the total NH$_3$ emissions in the U.S are derived from vehicles, with almost all the remaining NH$_3$ coming



from agricultural processes (Kean et al., 2009). Based on a "bottom-up" methodology, a city-specific

non-agricultural $NH_3$ emission inventory for 113 Chinese cities was recently established (Chang, 2014),

in which traffic (32.2%) was identified as the largest $NH_3$ emission source. At the global scale,

non-agricultural $NH_3$ emissions are one or two orders of magnitude smaller than the gross flux of gaseous

$NH_3$ between the Earth's surface and the atmosphere, which totals more than 50 Tg y$^{-1}$ (Schlesinger and

Hartley, 1992). Non-agricultural activities, however, are highly concentrated in urban areas and,

therefore, could be supposed to be significant sources of $NH_3$ in cities (Chang et al., 2012). Given the

important role of urban $NH_3$ emissions to form $PM_{2.5}$, $NH_3$ emission reduction has been regarded as the

key to curb severe haze pollution in Chinese mega-cities (Ye et al., 2011; Wang et al., 2011; Wang et

al., 2013).

Although isotopic techniques have been extensively accepted as a useful tool for source apportionment

of gases and PM (e.g., Cao et al., 2011; Felix et al., 2012; Liu et al., 2014b; Rudolph et al., 1997; Wang

et al., 2016; Xiao et al., 2012; Xiao et al., 2015; Xiao and Liu, 2002), there have been few studies to

date in terms of directly observing or quantifying the contribution of non-agricultural $NH_3$ in the

atmosphere (Felix et al., 2014; Liu et al., 2008). Greater scientific attention and regulatory efforts have

been giving to nitrogen oxides ($NO_x=NO+NO_2$) (Felix et al., 2012; Michalski et al., 2014; Walters and

Michalski, 2015; Walters et al., 2016; Walters et al., 2015) and sulfur dioxide ($SO_2$) (Barros et al., 2015;

Giesemann et al., 1994; Habicht and Canfield, 1997; Zhelezinskaia et al., 2014). Also, the conventional

method for analyzing $\delta^{15}N$, using elemental analyzer (EA) combustion with isotope-ratio mass

spectrometry (IRMS), normally requires more than 20 µg N for a single solid sample, which poses a

considerable challenge for passive sampling devices (Skinner et al., 2006). To overcome this technical

restriction, a landmark paper was published by Felix et al. (2013), in which they tried to combine the

bromate ($BrO^-$) oxidation of $NH_4^+$ to $NO_2^-$ with microbial denitrifier methods (bacteria converts $NO_2^-$ to

$N_2O$) to permit the N isotopic analysis of low concentration $NH_4^+$ passive samples, featuring the high

throughput of sample mass and low toxicity of reagents. However, the microbial denitrifier they used

needs careful cultivation and maintenance, which is time-consuming and also may present a challenge

for many isotope laboratories. Recently, a novel and robust chemical method for $\delta^{15}N\text{-}NH_4^+$ at natural

abundance has been developed (Liu et al., 2014a), which has major advantages over previous

approaches: (i) substantially simplified preparation procedures and reduced preparation time particularly





compared to the methods in which diffusion or distillation is involved since all reactions occur in the same vial and separation of $NH_4^+$ from solution is not required; and (ii) greater suitability for low volume

samples including those with low N concentration, having a blank size of 0.6 to 2 nmol.

The 2014 Asia-Pacific Economic Cooperation (APEC) summit, another major international event after the 2008 Olympic Games, was hosted in Beijing on 3-12, November. To ensure good air quality and traffic flow during the APEC Summit, a set of stringent measures to control atmospheric pollutants, including regulating vehicle travel (restricting traffic based on the odd and even number plate rule),

delaying winter heating (for a week), suspending coal-based industries and closing construction sites, were implemented in Beijing and its neighboring provinces over a month before and during the APEC summit (Chen et al., 2015; Li et al., 2015a; Tang et al., 2015; Xu et al., 2015b). This provided a unique city-wide experiment to isotopically examine the response of various $NH_3$ sources to such comprehensive and intensive mitigation efforts. In the present study, the isotopic signatures of various

$NH_3$ sources in China were determined for the first time. Moreover, the ambient $NH_3$ concentrations and their isotopic compositions were investigated before, during and after the APEC control period in Beijing. Based on the isotopic signatures of major sources we developed, a stable isotope mixing model was used to quantify the contributions of each $NH_3$ source so as to examine the effect of the control measures.

**2 Methodology**

**2.1 Ambient $NH_3$ Monitoring**

The Adapted Low-Cost Passive High Absorption (ALPHA) samplers (Centre for Ecology and Hydrology, Edinburgh, UK), one of the most widely recognized passive sampling devices (PSDs) (Puchalski et al., 2011; Tang et al., 2001; Xu et al., 2015a), were used to collect ambient $NH_3$ in this

study. The ALPHA sampler is a circular polyethylene vial (26 mm height, 27 mm diameter) with one open end. The vial contains a position for a 25 mm phosphorous acid-impregnated filter and a PTFE membrane for gaseous $NH_3$ diffusion. In the current study, triplicate ALPHA filters were used to collect $NH_3$ for IRMS at weekends (from Saturday to Monday) or weekdays (from Monday to Saturday) on the roof of a 4-floor building (12 m a.g.l) in the campus of the China Agricultural University (CAU),

Beijing (116.289°E, 40.032°N) before (18th-20th-25th-27th October-1st-3rd November), during





(3$^{rd}$-8$^{th}$-10$^{th}$-15$^{th}$ November), and after (15$^{th}$-17$^{th}$-22$^{nd}$-24$^{th}$-29$^{th}$ November) the APEC summit (normally at 8:00 local time). The site of the CAU campus represents a general urban background with relatively low-rise buildings in the surrounding area, ca. 16.3 km Northwest of Tiananmen Square (Fig. 1). In this study, the three filters of one sampling event were combined for a single analysis. Hourly mass

concentrations of gases (CO, SO$_2$, NO$_x$ and O$_3$) and PM$_{2.5}$ in Beijing are averaged from the data obtained from the 14 state-controlled environmental monitoring stations across the city (http://datacenter.mep.gov.cn/).

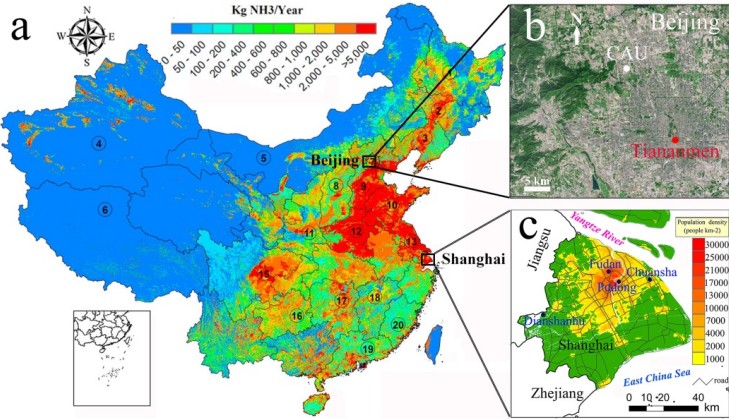

**Figure 1**.  (**a**) 1 km×1 km gridded NH$_3$ emission inventory in China in 2006 (modified from Huang et

al., (2012). (**b**) Location of the sampling site (China Agricultural University or CAU) in this study and

its spatial position relative to Tiananmen Square (Beijing urban center). (**c**) Atmospheric NH$_3$

monitoring network in Shanghai.

**2.2 NH$_3$ Source Sampling**

The Ogawa passive sampling device (Ogawa & Co., FL, USA) is another popular PSD that has been

successfully applied in the U.S. (Butler et al., 2015; Puchalski et al., 2011) and China (Chang et al., 2015; Meng et al., 2011) to determine the time-integrated NH$_3$ concentrations. The Ogawa PSD is a double-sided passive diffusion sampler equipped with a diffusive end cap, followed by a stainless-steel screen, and a 14mm quartz filter impregnated with phosphoric acid by the manufacturer. In this study, the Ogawa PSDs were used to collect NH$_3$ emitted from sources for isotopic analysis.





Six $NH_3$ emission sources were involved in this study, i.e., livestock (two pig sties), fertilizer

volatilization (laboratory simulation of $NH_3$ volatilization from urea-fertilized soil), human excreta

(septic tanks of a residential building and a teaching building), waste water (sewage water treatment

plant), solid waste (municipal waste transfer stations in a residential community and an educational

area), and vehicle (a heavily used urban tunnel) (Table 1 and SI Table S1). $NH_3$ emitted from fertilizer

volatilization was collected in the laboratory of Fudan University. To minimize the mixture of ambient

atmospheric $NH_3/NH_4^+$ (see Fig. 2 and SI Fig. S1 for two examples) and to examine the potential

influence of the macro environment, non-laboratory samples were collected within a confined space in

short periods (e.g., several hours) during both the warm (summer) and cold seasons (winter) between

June 2014 and January 2015. Descriptions of the sampling processes are detailed in SI Table S1.

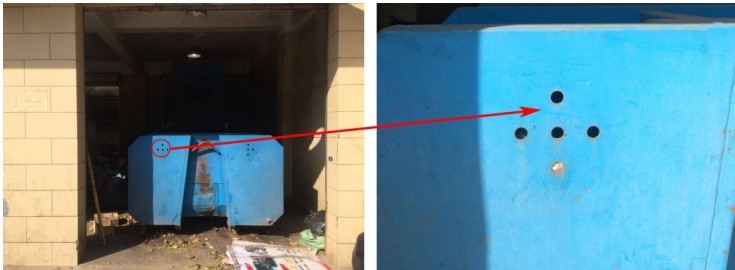


**Figure 2**. Field photos of a solid waste container in the solid waste transfer station we used for source

sampling at Fudan University. Before sampling, an Ogawa passive sampler for $NH_3$ is attached on a

plastic strip (1.5 m in length). The sampler is fitted into the container through the holes (shown on the

right panel).

The ambient average $NH_3$ concentration was less than 6 μg m$^{-3}$ (Chang et al., 2016), which was

between 1 and 3 orders of magnitude smaller than all emission sources we investigated (Table 1).

Therefore, the interference of ambient air to all source samples is not a major concern in our study.

HONO has been measured in significant concentrations in the UK and interfered with measurements of

$HNO_3$ by DELTA active samplers causing the instrument now to be coated with NaCl to avoid this

interference with $HNO_3$ measurements. The Ogawa filters used for trapping $NH_3$ were impregnated

with citric acid. Using a Discrete Auto Analyzer (Smartchem 200, AMS, Italy; the detection limit for

$NO_2^-$-N is 0.002 mg L$^{-1}$), we recently analyzed nearly 100 passive filter samples collected from our

passive ammonia network in Shanghai (Fig. 1). No detectable nitrite was found for almost all these





samples (including samples collected from a busy road tunnel (Chang et al., 2016), which provides

compelling evidence that passive collection of ammonia don't introduce interference of nitrite in this

study.

In our pilot study, the sampling period of each source was optimized to avoid insufficiency of $NH_3$

absorption, but more importantly, to minimize potential effects of N isotope fractionation. Taking the

source of human excreta for example, the concentrations of $NH_3$ in the exhausts of 15 ceiling ducts

from collecting septic tanks in 13 buildings with 6 functions ($2809^{+5803}_{-2661}$ µg m$^{-3}$) were higher than those

in ambient air by 3 orders of magnitude (Chang et al., 2015). Therefore, the insufficiency of $NH_3$

absorption is of no significant concern. However, the $\delta^{15}$N-$NH_3$ values of daily samples varied widely

(±10 per mil), suggesting that the isotope fractionation may occur during the process of

sampling/storage. After many tests by trial and error, we found that a sampling period of 2 hours could

provide sufficient N-$NH_3$ as well as avoid potential fractionation. Another example is the on-road

traffic source. The $\delta^{15}$N-$NH_3$ values of weekly (-11.9‰; -11.2‰), semimonthly (-11.7‰; -12.5‰) and

monthly samples (-12.0±1.8‰; n=4) in the exit of Handan tunnel were almost identical, suggesting that

passive $NH_3$ collection is an effective approach for isotopic analysis procedures.

### 2.3 Stable Isotope Analysis

A newly developed chemical method for $\delta^{15}$N-$NH_4^+$ of low $NH_4^+$ samples was used in the current study.

The detailed analytical procedures are given elsewhere (Liu et al., 2014a). Briefly, this method is based

on the isotopic analysis of $N_2O$, which is much less abundant in the atmosphere than $N_2$ and thus causes

minimal atmospheric contamination. Filter samples were firstly soaked with ultra-pure water (18.2

MΩ.cm). Concentrations of $NH_4^+$ were then analyzed using an ion chromatographic system (883 Basic

IC plus, Metrohm Co., Switzerland) equipped with a Metrosept C4/4.0 cation column. The eluent was

1.0 mM $HNO_3$ + 0.5 mM PDA. The detection limit for $NH_4^+$ was 0.0028 mg L$^{-1}$. After the

measurement of the $NH_4^+$ concentration, $NH_4^+$ in every sample was initially oxidized to $NO_2^-$ by

hypobromite (BrO$^-$) in a vial. $NO_2^-$ was then quantitatively converted into $N_2O$ by hydroxylamine

($NH_2OH$) under strongly acid conditions. The produced $N_2O$ was analyzed by a purge and cryogenic

trap system (Gilson GX-271, IsoPrime Ltd., Cheadle Hulme, UK) coupled to an IRMS (PT-IRMS)





(IsoPrime 100, IsoPrime Ltd., Cheadle Hulme, UK) at the *Stable Isotope Ecology Laboratory of Institute of Applied Ecology, Chinese Academy of Sciences*.

Isotope ratio values are reported in parts per thousand relative to atmospheric $N_2$ as follows:

$$\delta^{15}N\,(\text{‰}) = \frac{\left(^{15}N/^{14}N\right)_{sample} - \left(^{15}N/^{14}N\right)_{N_2}}{\left(^{15}N/^{14}N\right)_{N_2}} \times 1000 \quad (1)$$

Three international $NH_4^+$ standards (IAEA N1, USGS 25, and USGS26 with $\delta^{15}N$ values of +0.4‰, -30.4‰ and +53.7‰, respectively) were used to correct for the reagent blank and drift during isotope analysis of the produced $N_2O$. The standard deviation of $\delta^{15}N$ measurements is less than 0.3‰. Moreover, to enhance our confidence in the results determined by the PT-IRMS method, the $\delta^{15}N$ values of fertilizer-, vehicle-, human excreta-, and solid waste-derived $NH_3$ were also examined by the

EA-IRMS method at the *Shanghai Institutes of Life Sciences*, *Chinese Academy of Sciences* (SI Text S1).

**2.4 Isotope Mixing Model**

A stable isotope mixing models offer a statistical framework to estimate the relative contributions of multiple sources to a mixture, such as food-web structure, plant water use, air pollution, and many

other environments (Cole et al., 2011; Dai et al., 2015; Jautzy et al., 2015; Wang et al., 2016). A common problem, however, is having too many possible sources relative to isotopes to allow unique linear mixing solutions based on mass balance equations. To this end, Phillips and Gregg (2003) developed the model IsoSource, which solves iteratively for feasible mixing solutions, and has been well tested in numerous studies (e.g., Cole et al., 2011; Dai et al., 2015; Jautzy et al., 2015; Wang et al.,

2016). The model does not generate exact values for proportional contributions of each source, but instead provides a range of possible contributions or feasible solutions. The IsoSource addresses every possible combination of source proportions (summing to 100%) incrementally (e.g., 1%), then calculates the predicted isotope value for each combination using linear mass-balance equations. These predicted values are then examined to determine which ones fall within some tolerance range (typically

0.1‰) of the observed consumer isotope value, and all of these feasible solutions are recorded. The IsoSource model is available at https://www.epa.gov/eco-research/stable-isotope-mixing-models-estimating-source-proportions.



Considering the large number of possible sources for ambient $NH_3$, we use multiple lines of evidence (prior information) to constrain the emission sources in the mixing model analysis: (1) there was no

crop harvest activity in the North China Plain (NCP) during the APEC summit in Beijing. Besides, harvesting forests for fuelwood and timber has nearly disappeared in Beijing. Therefore, the contribution of biomass burning is considered minimal; (2) Beijing is 150 km inland from the Bohai Sea (the nearest sea) via Tianjin Municipality in the southeast. Therefore, marine source in Beijing can be neglected; and (3) previous work indicated that miscellaneous $NH_3$ sources like pets and household

products are minor $NH_3$ emissions in Beijing urban areas (Chang, 2014), which thus can be largely neglected.

In conclusion, ambient $NH_3$ in Beijing during our study period has been shown to be due to four main sources: livestock production, N-fertilizer application, on-road traffic emissions, and waste-derived emissions. The $\delta^{15}N$ average values for these four $NH_3$ emission sources will be built and served as the

baseline input to the IsoSource. For the traffic source, given that the relatively larger difference in terms of their $\delta^{15}N$ values in different seasons (see Table 1), the wintertime average value of $\delta^{15}N$ was used in this study because the APEC summit was held during winter. In brief, the N isotopic signatures for the sources of waste, livestock, traffic, and fertilizer are set as -37.8‰, -29.1‰, -16.5‰, and -50.0‰, respectively (Table 1 and SI Table S1). The source increment and mass balance tolerance

parameter values of 1% and 0.1‰, respectively, were applied. Model output files include all the feasible source combinations, with histograms and descriptive statistics on the distributions for each source. Results are expressed as box-and-whisker plots for these distributions.

### 3. Results and Discussion

### 3.1 Isotopic Signatures of $NH_3$ Emission Sources

Using N isotope as a tool to discriminate the contribution of various sources to ambient $NH_3$ concentration requires (i) well-established N isotopic compositions of $NH_3$ emission sources and (ii) well-constrained N isotope fractionation to allow separating different sources. In total 44 $NH_3$ source samples in this study, $\delta^{15}N$ values and $NH_3$ concentrations of these samples ranged from -52.0 to -9.6‰, and 33 to 6211 µg m⁻³, respectively (all data are presented in SI Table S1). These $NH_3$ sources can be

classified into four categories, i.e., fertilizer, livestock, traffic, and waste. For most sources, there was



no significant difference in terms of their $\delta^{15}$N values in different seasons (Table 1), indicating the effectiveness of our sampling strategy. The work of Felix et al. (2013) addressed some of the same issues and we found and reported similar results for the range of $\delta^{15}$N-NH$_3$ values (-56.1 to -2.2‰) from major NH$_3$ emission sources (including livestock, marine, vehicle, and fertilizer sources) as those

presented here. These two independent studies determining the $\delta^{15}$N values of major NH$_3$ sources arrive at the same conclusion: NH$_3$ emitted from volatilized sources has relatively low $\delta^{15}$N values, allowing them to be distinctly differentiated from NH$_3$ emitted from traffic sources that are characterized by relatively high $\delta^{15}$N values.

**Table 1.** $\delta^{15}$N-NH$_3$ values of ammonia sources, source location, and summer-winter comparison (mean

± SD).

| Category | Sub-category | Location | Season | NH$_3$ conc.($\mu$g m$^{-3}$) | $\delta^{15}$N-NH$_3$ (‰) | N |
|---|---|---|---|---|---|---|
| Livestock | pig | Shanghai pig farm | summer | 1329.6±175.8 | -30.3±1.3 | 3 |
| | | | winter | 586.6±113.2 | -28.2±1.5 | 4 |
| Traffic | on-road vehicle | Handan Tunnel | summer | 85.2±6.2 | -12.0±1.8 | 4 |
| | | | winter | 46.7±14.3 | -16.5±1.1 | 4 |
| Waste | solid waste | municipal waste transfer stations | summer | 443.4±92.6 | -31.1±1.0 | 4 |
| | | | winter | 315.0±48.5 | -36.6±0.9 | 4 |
| | wastewater | wastewater treatment plant | summer | 246.3±9.3 | -41.3±0.7 | 4 |
| | | | winter | 143.8±12.3 | -40.7±1.1 | 4 |
| | human excreta | septic tanks | summer | 4578.4±1400.3 | -38.4±0.9 | 4 |
| | | | winter | 4440.1±1288.6 | -38.6±1.0 | 4 |
| Fertilizer | urea | Fudan lab | / | 396.3±199.9 | -50.0±1.8 | 5 |

Road tunnels are excellent locations to provide emissions from a large number of vehicles mixed with SCRs and TWCs under 'real world conditions' (Liu et al., 2014c). Here we assume that the $\delta^{15}$N-NH$_3$ values collected from a heavily used tunnel like the Handan tunnel (around 120000 vehicles passing per day; an average of 65.9 $\mu$g m$^{-3}$ NH$_3$ or 12.4 times the levels of open environment (Chang et al.,

2016) can be accepted as the isotopic signatures of vehicles in China. Nevertheless, we note the $\delta^{15}$N-NH$_3$ values of vehicle exhausts that we collected from the Handan tunnel (-17.8 to -9.6‰; n=8)



are lower than the $\delta^{15}$N-NH$_3$ values determined by Felix et al. (2013) in the Squirrel Hill Tunnel, Pittsburgh (-4.6‰ and -2.2‰; n=2). This Sino-US difference may be partially attributed to a higher usage of TWCs in the traffic in the US.

The $\delta^{15}$N values of NH$_3$ from livestock (-31.7 to -27.1‰) and fertilizer (-52.0 to -47.6‰) that we measured are slightly lower than the range of $\delta^{15}$N-NH$_3$ values collected monthly from two dairy barns (-28.5 to -22.8‰) and a cornfield treated with urea-ammonia-nitrate fertilizer (-48.0 to -36.3‰) by Felix et al. (2013). These ranges of $\delta^{15}$N values are a function of the initial $\delta^{15}$N values of animal waste and fertilizer and variations in the bacteria populations, as well as other factors (temperature, wind, pH

etc.) that influence kinetic fractionation rates associated with NH$_3$ volatilization. Long-term (30 days) monitoring of $\delta^{15}$N and NH$_3$ emissions of manure measured by Lee et al. (2011) indicated that the dynamics of N isotope fractionation may be complicating the usefulness of the isotope approach as a tool for estimating NH$_3$ emissions in field conditions. In this sense, the shorter sampling period in our work should reflect the essence of $\delta^{15}$N values of livestock- and fertilizer-derived NH$_3$.

As a normal metabolic process, the release of NH$_3$ from human excreta has been well documented. However, most emission inventories involving human excreta have focused on pit latrines in rural areas of developing and middle income countries. In urban China, human excreta are typically stored in a three-grille septic tank under the building before disposal. After a series of anaerobic decomposition processes, a substantial amount of NH$_3$ will be generated and emitted through a ceiling duct. In the

present study, the concentrations of NH$_3$ in the ceiling ducts (4509.3±1248.0 µg m$^{-3}$; n=8) outweigh those in the open air by 3 orders of magnitude, and the $\delta^{15}$N-NH$_3$ values are seasonally consistent (-38.4±0.9‰ in summer and -38.6±1.0‰ in winter; Table 1), suggesting that human excreta may be an important and consistent source of NH$_3$ in urban areas. These data suggest that emissions of NH$_3$ from human excreta for an urban population of ~21 million people in Shanghai contribute 1386 Mg NH$_3$

annually to the atmosphere within the city, which corresponds to 11.4% of the total NH$_3$ emissions in the Shanghai urban areas (Chang et al., 2015). The $\delta^{15}$N values of wastewater-originated NH$_3$ (-41.0±0.9‰; n=8) are close to that of human excreta and also show no seasonal variation. Sampling in a stable and closed physical environment may be responsible for such a small range of isotopic variation. However, although also sampled in a closed environment, the $\delta^{15}$N values of municipal solid



waste demonstrate a much greater variation (-37.6 to -29.9‰), which may be due to the variable composition of solid waste.

**3.2 Source Apportionment of Ambient NH$_3$ in Beijing**

Hourly observations of major air pollutants (including PM$_{2.5}$, NO$_x$, CO, SO$_2$ and O$_3$) in Beijing are show in SI Fig. S2. The meteorological differences (e.g., temperature and wind speed) for the three periods pre, during and post APEC are not significant, suggesting that emission reduction strategies implemented during the APEC summit were successful (SI Fig. S2). It should be noted that several control measures, i.e., closing factories within 200 km of the city center and stopping the entrance of out-of-city vehicles, had been undertaken in Beijing and its neighboring regions before the summit. Therefore, the before-during comparison of some pollutants like SO$_2$ are not in stark contrast in terms of their mass concentration (SI Fig. S2). The evolution of ambient NH$_3$ mass concentrations measured at CAU shows a similar pattern with CO and NO$_x$ (SI Fig. S3). Before the opening of the summit (from 18$^{th}$ October to 3$^{rd}$ November), NH$_3$ concentrations averaged 9.9 μg m$^{-3}$ and ranged from 6.9 to 11.0 μg m$^{-3}$. During the summit session (from 3$^{rd}$ to 15$^{th}$ November, the ending date of the summit is 13$^{th}$ November), this was reduced to 7.3 μg m$^{-3}$ with a range of 5.8 to 8.6 μg m$^{-3}$. After the APEC summit (from 15$^{th}$ to 29$^{th}$ November), the NH$_3$ concentration levels rebounded to an average of 12.7 μg m$^{-3}$ (ranging from 10.7 to 17.7 μg m$^{-3}$). In other words, the NH$_3$ concentrations were reduced by 20.0% during the APEC summit compared with the period before it. Compared with the period after control, the concentrations were 74.5% lower than that during the summit (Table 2).

On the basis of the $\delta^{15}$N values of NH$_3$ emission sources and ambient $\delta^{15}$N-NH$_3$ samples (Table 2), the ranges (within 5 and 95 percentiles) of relative contribution fractions of each NH$_3$ source to the ambient atmosphere were modeled by the IsoSource and depicted in Fig. 3a-d. Of these, sources of traffic and fertilizer are better constrained than livestock and waste. This is because the $\delta^{15}$N-NH$_3$ values of the ambient atmosphere are closer to the latter two. For example, after the APEC summit in our study period, the $\delta^{15}$N-NH$_3$ values of the ambient atmosphere averaged -30.7‰, which is very close to the isotopic signature of livestock (-29.1‰), thus leading to weaker constraint with 5 and 95 percentiles ranges from near 0 to 0.7 (Fig. 3).





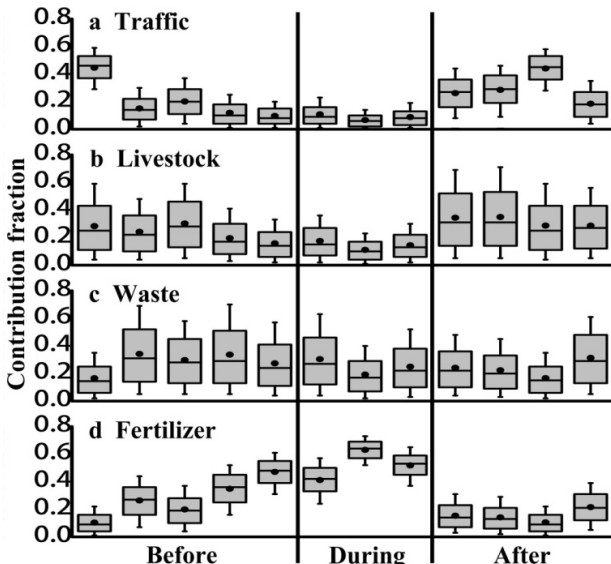

**Figure 3.** Time series of the box-whisker plots of the contribution fraction of ambient $NH_3$ from different sources estimated from the IsoSource model. The box boundaries represent the 25th and 75th percentile; the horizontal line is the median, and the whiskers mark the 10th and 90th percentiles. The dots denote the mean values of the distribution.

For each sample, information regarding the $\delta^{15}N$ value, the $NH_3$ concentration and its source contributions is listed in Table 2. Fig. 4 illustrates the overall source contribution proportions before, during and after the summit, in which traffic, waste, livestock, and fertilizer comprised 18.3, 27.1, 24.0, and 30.6% of the whole period (Table 2), respectively. Specifically, the contribution of traffic initially decreased by 58.7% and then doubled, and this represents the largest change among the four sources (Table 2). Considering the absolute contributions of different sources to ambient $NH_3$, these results show that traffic is the most sensitive source to the emission control measures. This result was expected since over half the vehicles in Beijing were banned from entering the city during the APEC summit (Note that before the summit, out-of-city vehicles had already been banned from entering Beijing). Based on the vehicle $NH_3$ emission factor obtained from real-world tunnel tests, Liu et al. (2014c) and Chang et al. (2016) recently reported that vehicles contribute 8.1% and 12.0% to total $NH_3$ emissions in the Pearl River Delta region and Shanghai urban areas, respectively. With a total of 5.5 million vehicles in 2014, the traffic fleet in Beijing was far ahead of other cities. Thus vehicle sources might contribute more than 20% of the total $NH_3$ to the air within Beijing.



**Table 2**. $\delta^{15}$N values, NH$_3$ concentrations and source contributions for all samples collected before, during, and after the APEC summit in Beijing.

| Sample ID | $\delta^{15}$N (‰) | NH$_3$ conc. (μg m$^{-3}$) | Absolute contribution of NH$_3$ conc. (μg m$^{-3}$) | | | | Relative contribution of NH$_3$ conc. (%) | | | |
|---|---|---|---|---|---|---|---|---|---|---|
| | | | Traffic | Waste | Livestock | Fertilizer | Traffic | Waste | Livestock | Fertilizer |
| Before-1 | -27.1 | 11.0 | 2.0 | 3.4 | 3.1 | 2.4 | 18.5 | 31.3 | 28.6 | 21.6 |
| Before-2 | -35.7 | 8.9 | 1.3 | 3.1 | 2.2 | 2.4 | 15.1 | 34.2 | 24.3 | 26.4 |
| Before-3 | -33.3 | 10.2 | 2.1 | 3.0 | 3.1 | 2.0 | 20.2 | 29.7 | 30.2 | 19.9 |
| Before-4 | -37.8 | 8.6 | 1.0 | 2.9 | 1.7 | 3.0 | 12.0 | 33.6 | 19.6 | 34.8 |
| Before-5 | -40.1 | 6.9 | 0.7 | 1.9 | 1.1 | 3.3 | 9.7 | 27.4 | 15.8 | 47.1 |
| During-1 | -39.0 | 8.6 | 0.9 | 2.6 | 1.5 | 3.5 | 10.8 | 30.4 | 17.5 | 41.2 |
| During-2 | -43.1 | 5.8 | 0.4 | 1.1 | 0.6 | 3.7 | 6.7 | 19.2 | 11.1 | 63.1 |
| During-3 | -41.0 | 7.5 | 0.7 | 1.9 | 1.1 | 3.9 | 8.8 | 25.0 | 14.4 | 51.8 |
| After-1 | -31.1 | 11.6 | 3.0 | 2.8 | 4.0 | 1.8 | 26.2 | 24.1 | 34.4 | 15.3 |
| After-2 | -30.4 | 17.7 | 5.1 | 4.0 | 6.2 | 2.5 | 28.6 | 22.4 | 34.9 | 14.1 |
| After-3 | -27.2 | 10.7 | 4.7 | 1.8 | 3.1 | 1.1 | 44.1 | 16.7 | 28.7 | 10.5 |
| After-4 | -34.0 | 11.0 | 2.0 | 3.4 | 3.2 | 2.4 | 18.5 | 31.3 | 28.6 | 21.6 |
| Overall | -35.0±5.4 | 9.9±3.1 | 2.0±1.6 | 2.7±0.8 | 2.6±1.6 | 2.7±0.8 | 18.3±10.6 | 27.1±5.7 | 24.0±8.1 | 30.6±16.8 |
| Before | -34.8 | 9.1 | 1.4 | 2.9 | 2.2 | 2.6 | 15.1 | 31.2 | 23.7 | 30.0 |
| During | -41.1 | 7.3 | 0.7 | 1.9 | 1.1 | 3.7 | 8.8 | 24.9 | 14.3 | 52.0 |
| After | -30.7 | 12.7 | 3.7 | 3.0 | 4.1 | 1.9 | 29.4 | 23.6 | 31.7 | 15.4 |
| Change (%) During *VS*. Before | 17.9↓ | 20.0↓ | 53.8↓ | 34.7↓ | 51.8↓ | 42.2↑ | 58.7↓ | 0.9↓ | 41.0↓ | 87.6↑ |
| Change (%) After *VS*. During | 25.3↑ | 74.5↑ | 462.7↑ | 60.5↑ | 280.6↑ | 47.5↓ | 234.2↑ | 5.0↓ | 120.8↑ | 70.5↓ |

The waste-originated percentages in Fig. 3 and 4 remain fairly constant over the sampling period, appearing to be a stable and important NH$_3$ contributor in Beijing. Compared with wastewater and solid

water, NH$_3$ emissions from human excreta through *in situ* sceptic tank system are far from quantified. Based on an extensive measurement campaign, we estimated that the population of ~21 million people living in the urban areas of Shanghai annually emitted approximately 1386 Mg NH$_3$, which corresponds to over 11.4% of the total NH$_3$ emissions in the urban areas (Chang et al., 2015). Non-agricultural sources-merged with waste and traffic NH$_3$ emissions-collectively account for nearly

50% of ambient NH$_3$ before and after the APEC summit, which cannot be explained by previous work of emission inventories (e.g., Fu et al., 2013; Huang et al., 2011; Huang et al., 2012; Kang et al., 2016; Li et al., 2015b; Zhang et al., 2009; Zhang et al., 2010). Our results do not contradict a commonly held belief that agriculture is responsible for the vast majority (normally >90%) of total ammonia emissions at a regional scale. However, the results show that within urban areas, non-agricultural sources are very





important. A consequence of the new results is that measures to improve air quality in urban areas of

China need to include measures to reduce both agricultural and non-agricultural sources of $NH_3$.

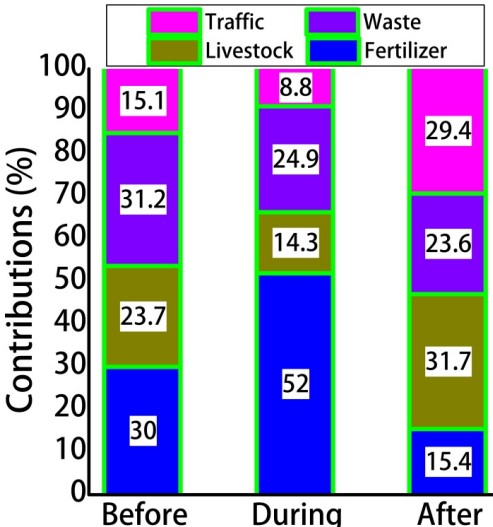

**Figure 4**. The overall contribution proportion (%) of the four sources to ambient $NH_3$ identified by the
IsoSource before, during, and after the APEC summit in Beijing.

The North China Plain (NCP) is one of the most intensive agricultural regions in China, enjoying a

good reputation of "China's granary" (Ju et al., 2009). Situated on the northern edge of the NCP with

mountains to the North and West, Beijing is a receptor of agricultural $NH_3$ from rural areas. During our

study period, crops in the NCP had been harvested and thus fertilizer application would have been very

limited. However, our results show that fertilizer is the largest contributor, accounting for 30.6%

throughout the sampling periods (Table 2). One explanation might be the prevalence of intensive urban

agricultural production with high nitrogen fertilizer input in the suburban areas. Beijing's increase in

land area from 4822 km² in 1956 to 16808 km² in 1958 led to the increased adoption of peri-urban

agriculture. Such "suburban agriculture" contributed ~70% of non-staple food in Beijing, mainly

consisting of vegetables and milk, to be produced by the city itself in the 1960s and 1970s (Cai, 2003).

In the late 1990s, recognizing the importance of urban agriculture to sustainable urban development,

Beijing's municipal government launched an official program encouraging multi-function urban

agriculture in peri-urban areas by supporting the development of "agro-parks", which not only produce

food but also attract tourism and are used as educational tools (Cai, 2003). One of the recent



experiments in urban agriculture is the Modern Agricultural Science Demonstration Park in
Xiaotangshan Town, Changping District (Cai, 2003). Today, Beijing is leading the way in using
smart-city technologies to make urban farming more sustainable. In addition to suburban agriculture,
there are 17 golf courses with 2280 ha. greens in Beijing; some of them are located at the urban areas
(Chang, 2014). The turf grass of golf course typically needs 200-400 kg N ha$^{-1}$ yr$^{-1}$ as N fertilizer to
achieve high performance (Wong et al., 1998; Wong et al., 2002; Zhang, 2002), which should be
considered as an overlooked NH$_3$ contributor in Beijing.

### 3.3 Limitations and Outlook

The dataset reported in this study represents, to the best of our knowledge, the first attempt to partition
urban atmospheric NH$_3$ sources. Considering the current nascent stage of partitioning NH$_3$ sources
using stable isotope approach, there are several unsolved problems that could potentially undermine the
above-mentioned results. One of our fundamental assumptions in this study is that the measured NH$_3$
was directly from NH$_3$ emission sources. In other words, we treated the measured NH$_3$ as the mixture
of primary NH$_3$ sources without "gas-aerosol conversion" fractionation. But in fact, NH$_3$↔NH$_4^+$
equilibrium will cause $^{14}$N to be preferentially associated with NH$_3$ and $^{15}$N to be enriched in NH$_4^+$ of
PM due to the stronger associative strength of $^{15}$N than $^{14}$N in NH$_4^+$ (Kawashima and Kurahashi, 2011;
Yeatman et al., 2001). In Beijing, earlier studies confirmed that the atmosphere in Beijing was
NH$_3$-limited, suggesting that acidic gases could not be fully neutralized to form ammonium salts
(Ianniello et al., 2010; Wang et al., 2016). Therefore, ammonium salts had much less opportunity to
volatilize to NH$_3$ to exert substantial isotopic effect through NH$_3$↔NH$_4^+$ equilibrium. Still, it is critical
to develop a controlled laboratory system to fundamentally understand the characteristics and
mechanisms of N isotope fractionations during the process of NH$_3$ transformation (Li et al., 2012).

Several additional factors could introduce uncertainty in the solutions of isotope mixing model. Given
the complexity of urban NH$_3$ sources, no definitive solution exists in a linear mixing model with one
isotope system tracer ($\delta^{15}$N) in the current study. Recommended future studies should include the
combinations of different types of isotope ratio measurements and the adoption of more sophisticated
Bayesian mixing models. The isotopic signature of sources like on-road traffic still remains uncertain.
Some may argue that since NH$_3$ (also NO$_x$) is known to be a component of vehicle-emitted exhaust,



why not collect vehicle-emitted $NH_3$ directly from the tailpipes. To our knowledge the $\delta^{15}N$ of vehicle-emitted $NH_3$ has not previously been assessed. However, a recent research paper from Walters et al. (2015) addressing the $\delta^{15}N$ of vehicle-emitted $NO_x$ may shed some light on this issue. In that

paper, the $\delta^{15}N$-values of $NO_x$ emitted from 26 different vehicles ranged from -19.1‰ to +9.8‰, much higher than the variation of $\delta^{15}N$-$NH_3$ collected from the Handan tunnel in our research. In road diesel Selective Catalytic Reduction (SCR) applications, a urea-in-water solution is used as the reduction agent. Urea is injected in the exhaust line and is decomposed over a catalyst to $NH_3$. In this case, the $\delta^{15}N$ values of vehicle-emitted $NO_x$ and $NH_3$ can hardly be the same. However, as a direct product of

NO reduction on the catalyst surface of TWCs ($2NO+5H_2 \rightarrow 2NH_3+2H_2O$ and/or $2NO+2CO+3H_2 \rightarrow 2NH_3+2CO_2$), $NH_3$ emitted from light-duty vehicle exhausts can be expected to have similar $\delta^{15}N$-values to vehicle-emitted $NO_x$. In this regard, the tunnel test has a unique advantage in measuring the overall isotopic signatures of vehicle-emitted pollutants. Therefore, we believe that the $\delta^{15}N$-$NH_3$ values of the samples collected from Handan tunnel in this study are representative as the

isotopic signatures of vehicles in China.

Despite the potential limitations in this study, given the importance of $NH_3$ to $PM_{2.5}$ formation, this work can be expected to enrich the discussion on the methodologies (including stable isotope analysis) in terms of identifying the largest $NH_3$ sources in urban atmosphere where policy efforts relating to emissions abatement can be directed to deliver the largest impact.

**4. Conclusions**

Firstly, we establish a pool of isotopic signatures ($\delta^{15}N$-$NH_3$) for the major $NH_3$ emission sources in China. The $\delta^{15}N$-$NH_3$ source inventory confirms that $NH_3$ emitted from on-road traffic has much higher $\delta^{15}N$ values (-14.2±2.8‰), allowing them to be differentiated from other sources, such as livestock (-29.1±1.7‰), waste (-37.8±3.6‰), and fertilizer (-50.0±1.8‰).

Secondly, we demonstrated that the isotopic source signatures of $NH_3$ represent an emerging tool for partitioning $NH_3$ sources. Taking advantage of the implementation of stringent air quality control measures during the APEC summit in Beijing, the IsoSource modeling results indicate that the overall contribution of traffic, waste, livestock, and fertilizer to ambient $NH_3$ mass concentrations is 18.3%,

27.1%, 24.0%, and 30.6%, respectively, in which traffic is the most sensitive to control measures. Our results clearly show that non-agricultural sources (traffic and waste) of NH$_3$ are of critical importance in megacities like Beijing. Therefore, in addition to current SO$_2$ and NO$_x$ controls, China also needs to allocate more scientific, technical, and legal resources on controlling non-agricultural NH$_3$ emissions in the future.

**Acknowledgements**

This work would have been impossible without the thought-provoking discussions with Prof. Yunting Fang at the Institute of Applied Ecology (IAE), Chinese Academy of Sciences. We sincerely thank Dr. Ying Tu at IAE, Mrs. Yue'e Yang and Mrs. Zi Li at the Shanghai Institutes of Life Sciences, for their assistance with the isotopic analysis. We also acknowledge the Qingyue Open Environmental Data Centre (data.epmap.org) for the unconditional help in terms of providing criterial pollutants monitoring 430 data. Yunhua Chang acknowledges the support of the Gao Tingyao Scholarship and the start-up grant for outstanding young scholars by the SUST. This work was partially funded by the National Natural Science Foundation of China (Grant Nos. 21377029, 21277030, 40425007, and 31421092) and the State Basic Research Program (2014CB954200).

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
