# Peer review of "Source apportionment of atmospheric ammonia before, during, and after the 2014 APEC summit in Beijing using stable nitrogen isotope signatures"

_Atmospheric Chemistry and Physics, 2016_

## Short Comment (SC1) · 25 Jun 2016

Dear readers, especially reviewers,

I would like to thank you for your valuable time in reviewing our MS.

All our modeling results for each sample were depicted in Figure 3, and their values were reported in Table 2. The modeling results show in Figure 3 are correct. However, I am deeply SORRY to tell you that I noticed a mistake in Table 2: the first sample (Before-1) and the last sample (After-4) had the same data in "Relative contribution of NH3 conc." for each NH3 source.

After double check, the data of the first sample in Tale 2 is wrong. The correct (wrong)

value for the relative contribution of traffic, waste, livestock, and fertilizer is 44.6% (18.5%), 16.6% (31.3%), 28.3% (28.6%), and 10.4% (21.6%), respectively.

Obviously, this mistake could underestimate the contribution of traffic to ambient NH3 before the APEC summit. However, I would like to emphasize that this mistake has no influence on our conclusion. Specifically, the wrong (correct) average contribution of traffic, waste, livestock, and fertilizer to ambient NH3 throughout our sampling period is 18.3% (20.4%), 27.1% (25.9%), 24.0% (24.0%), and 29.7% (30.6%), respectively.

I guess the reason of the mistake I made is because that the first sample (Before-1) and the last sample (After-4) have the same value in NH3 mass concentration, and I confused their modeling results during data processing. We are pleased to provide all data involved in this study as requested.

After communication with the handling editor, we've been told that we cannot make any corrections at the current stage and been suggested that we can make this statement as Authors' comments, and then make all the corrections in the final version of the paper (ACP version if it is accepted) together with other changes after addressing your concerns.

Sorry again for any inconvenience I've made.

Best Regards,

Yunhua Chang June 25, 2016 (the day of my PhD graduation at FD)

---

## Referee Comment (RC1) · Anonymous Referee #1 · 2 Aug 2016

The manuscript by Chang et al. studied the sources of atmospheric ammonia before, during, and after the 2014 APEC summit in Beijing using stable nitrogen isotope signatures coupled with an isotope mixing model. The source contributions of traffic, waste, livestock, and fertilizer were quantified and compared. The results showed a substantial decrease (58.7%) of traffic emission during the APEC period with strict emission controls. The results also showed that non-agricultural sources (traffic and waste) of NH3 play an important role in particle pollution in the megacity of Beijing, which has important implications for future air pollution mitigating strategies. This is a pioneer study by applying isotopic measurements into source apportionment of NH3. Such an approach could be very powerful in future source apportionment studies particularly if

combining with more collocated measurements. I recommend it for publication after addressing the following two comments.

Comments:

1. One of the assumptions of this study is that the contribution of biomass burning is considered minimal. The authors need to address the uncertainties of this assumption because Xu et al. (2015) showed that biomass burning contributed 12 – 19% to total organic aerosols before and during APEC in Beijing.

2. Coal combustion is the dominant source of aerosol particles during the heating season in Beijing. Unfortunately, such a source is missed in this study, which will affect the source apportionment results.

References:

Xu, W. Q., et al.: Aerosol composition, oxidation properties, and sources in Beijing: results from the 2014 Asia-Pacific Economic Cooperation summit study, Atmos. Chem. Phys., 15, 13681-13698, 10.5194/acp-15-13681-2015, 2015.

---

## Referee Comment (RC2) · Anonymous Referee #2 · 2 Aug 2016

Manuscript ID: acp 2016-432 Title: Source apportionment of atmospheric ammonia before, during, and after the 2014 APEC summit in Beijing using stable nitrogen isotope signatures The authors have presented data which increases the inventory of isotopic signatures of ammonia emission sources, an area lacking in data. The authors have characterized isotopic signatures of NH3 emissions sources in China, a region of the world where this has not been done, and the data agrees closely with data obtained in other areas of the world. This data can be employed by researchers to quantify NH3 emissions contributing to ambient atmosphere. This is valuable as many NH3 emission sources are nonpoint sources making them difficult to quantify. Using the nitrogen isotope signatures the authors have estimated urban source contribution before and

after a major event in Beijing in which air quality measures were employed during the event. The authors provide isotopic evidence that vehicles are a major contributor to urban NH3 concentrations and the isotopic signature in the city changed to reflect the reduction in vehicle emissions resulting from air quality measures. The estimation of other source contributions is likely confounded by potentially overlapping source signatures and this should be more adequately addressed by the authors. This a novel approach to assessing NH3 source contributions in an urban setting and the authors have presented the data in a clear and concise manner. If the major and minor issues below are addressed, I believe the manuscript could be accepted to ACP.

Major comments:

Source apportionment

The authors use an isotope mixing model to predict NH3 source apportionment. The endmember signatures used in the model are vehicles, fertilizer, livestock waste, and human waste. The authors provide evidence that the vehicle endmember signature is significantly different from the other three endmembers but the other endmembers signatures are similar and with more sampling it is likely that these signatures will overlap. This is because the three sources are essentially the same "volatilized NH3" source and if the literature data is taken into consideration, these sources' signatures are observed to overlap. I don't think the authors have a viable case for assigning the source signatures they present to each source. I do however believe they could combine the "volatilized NH3" sources as one endmember and vehicles as another endmember. While this would only provide insight to vehicle source apportionment, I think it is a more realistic approach. If the authors keep the mixing model as is, they need to explain the caveats associated with the estimates of the "volatilized NH3" sources and include error analysis. Rather than reporting specific values a range should be reported that represents the deviation and error involved in the calculations.

Vehicle sampling Unlike the US, a major urban NH3 source in China is human waste

which has been found to have a $\delta 15N$-NH3 value of $\sim$ -41 to -30 permil in this study. The vehicle exhaust in this study was sampled in a tunnel not directly from tailpipe. Most highly trafficked tunnels have ventilation systems that flush the tunnel with ambient air constantly. If the urban ambient air NH3 is mainly from waste and the tunnels are flushed with ambient air, this mixing would lower the $\delta 15N$ value of the NH3 sampled at the tunnel. Do these tunnels have ventilation systems? Could mixing with ambient air confound the $\delta 15N$ vehicle signal?

Line 119: All three filters from a sampling event were combined for single analysis. Why? Was there not enough N for analysis? This doesn't allow for reporting of the deviation, if any among samplers. Did the authors evaluate deviation among samplers? If so this should be included.

Line 166: When sampling exhaust from septic tanks the authors state "However, the $\delta 15N$-NH3 values of daily samples varied widely ($\pm 10$ per mil), suggesting that the isotope fractionation may occur during the process of sampling/storage. After many tests by trial and error, we found that a sampling period of 2 hours could provide sufficient N-NH3 as well as avoid potential fractionation" It seems the daily samples would be more representative of the source and the fractionation that is occurring is representative of the source. The magnitude of fractionation is going to change under varying environmental conditions but this is a symptom of the source type and should be evaluated as the range in source signature.

Line 284: The authors state "However, although also sampled in a closed environment, the $\delta 15N$ values of municipal solid waste demonstrate a much greater variation (-37.6 to -29.9‰, which may be due to the variable composition of solid waste" There are many more factors that may alter the fractionation during volatilization (e.g. pH, temperature, wind, moisture...) This again emphasizes the inability to use single source signatures for sources that are all a product of volatilized ammonia.

Line 400: The authors state "However, as a direct product of NO reduction on the catalyst surface of TWCs (2NO+5H2→2NH3+2H2O and/or 2NO+2CO+3H2→2NH3+2CO2), NH3 emitted from light-duty vehicle exhausts can be expected to have similar $\delta$15N-values to vehicle-emitted NOx." This reasoning is not sound. NO produced may have a different original $\delta$15N value then the NH3 being used in the TWC and the fractionation factor of the two different compounds caused by the TWC process could be very different. There is not valid evidence to state that the $\delta$15N of vehicle NOx and NH3 would be the same.

Minor comments Line 28: APEC should not be abbreviated Line 51: delete "extensive" Line 160: "don't" should be replaced with "doesn't" Line 198: "A" should be deleted Lin3 229: The wording "far ahead" is not the appropriate descriptor here and should be changed.

Figure 3: The x-axis labeling isn't sufficient. Are the boxes in order of sampling period? If so provide the timeframe on the axis

———————————————

---

## Author Comment (AC2) · 22 Aug 2016

We thank two referees for their careful considerations of the manuscript and their well thought-out comments. These certainly helped to significantly improve the paper. We've addressed all comments and questions below in the form of point-by-point responses. The referees' comments are in *Italic* and our responses are in normal font. Text changes in the revised manuscript are highlighted in color (including a mistake we mentioned in our short comments).

Referee 2:

*The authors have presented data which increases the inventory of isotopic signatures of ammonia emission sources, an area lacking in data. The authors have characterized isotopic signatures of $NH_3$ emissions sources in China, a region of the world where this has not been done, and the data agrees closely with data obtained in other areas of the world. This data can be employed by researchers to quantify $NH_3$ emissions contributing to ambient atmosphere. This is valuable as many $NH_3$ emission sources are nonpoint sources making them difficult to quantify. Using the nitrogen isotope signatures the authors have estimated urban source contribution before and after a major event in Beijing in which air quality measures were employed during the event. The authors provide isotopic evidence that vehicles are a major contributor to urban $NH_3$ concentrations and the isotopic signature in the city changed to reflect the reduction in vehicle emissions resulting from air quality measures. The estimation of other source contributions is likely confounded by potentially overlapping source signatures and this should be more adequately addressed by the authors. This a novel approach to assessing $NH_3$ source contributions in an urban setting and the authors have presented the data in a clear and concise manner. If the major and minor issues below are addressed, I believe the manuscript could be accepted to ACP.*

Thanks for the recognition of our contribution. Please check our point-by-point responses to the major and minor issues raised.

*The authors use an isotope mixing model to predict $NH_3$ source apportionment. The endmember signatures used in the model are vehicles, fertilizer, livestock waste, and human waste. The authors provide evidence that the vehicle endmember signature is significantly different from the other three endmembers but the other endmembers signatures are similar and with more sampling it is likely that these signatures will overlap. This is because the three sources are essentially the same "volatilized $NH_3$" source and if the literature data is taken into consideration, these sources' signatures are observed to overlap. I don't think the authors have a viable case for assigning the source signatures they present to each source. I do however believe they could combine the "volatilized $NH_3$" sources as one endmember and vehicles as another endmember. While this would only provide insight to vehicle source apportionment, I think it is a more realistic approach. If the authors keep the mixing model as is, they need to explain the caveats associated with the estimates of the "volatilized $NH_3$" sources and include error analysis. Rather than reporting specific values a range should be reported that represents the deviation and error involved in the calculations.*

(1) We agree with the reviewer's comments that some of the isotopic signatures of volatilized sources could overlap and this problem should be addressed properly.

We think the source classification in our study is reasonable.

Our NH$_3$ source classification (Figure below) is based on:

a. The difference of isotopic signatures.

In our MS we concluded that "NH$_3$ emitted from volatilized sources has relatively low $\delta^{15}$N values, allowing them to be distinctly differentiated from NH$_3$ emitted from traffic sources that are characterized by relatively high $\delta^{15}$N value". Obviously, traffic-derived and fertilizer-volatized NH$_3$ has the highest and the lowest $\delta^{15}$N values, respectively. However, "three sources are essentially the same volatilized NH$_3$" does not necessarily signify that all volatilized NH$_3$ has a similar variation range of $\delta^{15}$N-NH$_3$ values (data overlap), nor does it mean that these volatilized NH$_3$ sources should be classified as a single source category. Temperature is an important parameter, but not the only parameter in determining the differences of $\delta^{15}$N-NH$_3$ values among different NH$_3$ sources. In other words, even if data overlap may occur within volatilized sources, they can still be further classified as several sub-categories because of their fundamentally different emission mechanisms (e.g., organic N *vs.* inorganic N; see discussion later);

[Figure]

Variation range of isotopic signatures for different NH$_3$ sources in our study.

b. The results of emission inventories or the practical emission situation of our targeted study area.

If our sampling site is located in rural areas, then we agree with the reviewer that all volatilized sources can be combined as a single source because NH$_3$ source contributions in rural areas are much simpler: agricultural volatilized sources predominate the NH$_3$ emission budget. However, in urban areas, except for on-road traffic, there are many other important non-agricultural NH$_3$ sources (also volatized sources). Urban wastewater treatment plants and solid waste are arguably two of the most important NH$_3$ sources in urban China. As a case study, we've identified that human excreta stored in septic tanks in Shanghai is a stable and important source of atmospheric NH$_3$, contributing to over 11% of the total NH$_3$ emissions in the Shanghai urban areas. Therefore, it is of critical importance to incorporate the isotopic signatures

of waste-related NH₃ sources into the isotope mixing model.

More importantly, even in natural/rural areas, we still need to take account of emission inventory information. If we combined all volatilized sources as a single source, the underlying logic is that every volatilized source contributes equally to ambient NH₃ concentrations. However, wastewater and solid waste, generally, have an insignificant contribution to the rural NH₃ budget. Besides, **if we simply separate all NH₃ sources into two categories (volatilized source and combustion source) without the consideration of the emission inventory, then we are highly likely to get into a trouble: some $\delta^{15}N$ values of ambient NH₃ samples may be beyond the endmember, or the isotopic signatures of NH₃ sources.**

To facilitate our explanation, we would like to give a counterexample. In a recent paper entitled "Fossil fuel combustion-related emissions dominate atmospheric ammonia sources during severe haze episodes: Evidence from $^{15}N$-stable isotope in size-resolved aerosol ammonium", Pan *et al*. (2016) presented the isotopic measurements of size-resolved aerosols in Beijing, summarizing that fossil fuel-related ammonia emissions (including traffic, coal combustion and power plants NH₃ slip) have overtaken agricultural activities as the dominant source of atmospheric NH₃ during the hazy days of 2013.

There are three NH₃ sources, i.e., agricultural NH₃ volatilization, fossil fuel combustion, and power plant NH₃ slip, considered in Pan *et al*. (2016), and the average $\delta^{15}N$ values (specific values) of these sources were used as isotope signatures ($\delta^{15}N$-NH₃) to estimate their relative contributions to the ambient NH₃ in Beijing. We fully understand the author's consideration in terms of the classification of NH₃ sources: agricultural ammonia is emitted at environmental temperature, the process of fossil fuel combustion can directly emit ammonia at high temperature, and ammonia slipped from power plant is the residue of gaseous reductants (typically anhydrous ammonia, aqueous ammonia or urea) that are subjected to medium temperature, and the isotopic signatures of these sources can be separated based on various temperatures.

However, we have to point out that temperature-only-based NH₃ source classification has a few severe problems, which may lead to wrong conclusions and therefore could potentially mislead China's policy on future NH₃ emissions reduction.

It is well accepted that agricultural activities-fertilizer application merged with livestock production-are the largest contributors of NH₃ emissions at a regional or global scale. However, recent works reveal that NH₃ volatilized from fertilizer application and livestock waste have distinct $\delta^{15}N$ values, which can also be reflected by the large variation range of $\delta^{15}N$ values for agricultural source in Pan *et al*. (2016) (see Figure 3). This is because fertilizer application and livestock waste generally represent two totally different nitrogen forms, i.e., inorganic and organic nitrogen, respectively. Situated on the northern edge of the North China Plain, one of the most

intensive agricultural regions in China, Beijing is regarded as a receptor of agricultural ammonia from rural areas. In Pan *et al*. (2016), fertilizer application and livestock waste were combined as a single source (volatilized source), which could inevitably underestimate the contribution of agricultural activities to the ambient $NH_3$ in Beijing.

If temperature is a decisive factor, then power plant $NH_3$ slip considered as a major $NH_3$ source in Pan *et al*. (2016) should be reasonable. However, Pan *et al*. (2016) claimed that during haze periods, **49%** of $NH_3$ in the ambient atmosphere of Beijing was derived from power plant $NH_3$ slip, which can hardly be true. In September 2013, a five-year plan was introduced in Beijing to slash coal consumption, and there were only four coal-fired power plants (CFPP) operating near the city's urban areas during wintertime. In 2016, all CFPP in Beijing will be shuttled and replaced with gas-fired power plants to cut pollution. The replacement by the four gas-fired power plants will help cut emissions by 10000 t of sulfur dioxide and 19000 t of nitric oxide annually. Although ammonia slip is a common issue with SCR (Selective Catalytic Reduction) technology used in CFPP for removal of nitric oxide, the mass concentration of ammonia (typically 3-5 mg $NH_3$ $m^{-3}$) in flue gases is two or three orders of magnitude smaller than that of $NO_x$. Moreover, it is necessary to consider that although there are many CFPPs surrounding Beijing in the North China Plain, most of these are co-located with intensive agricultural production areas.

(2) We also fully agree with the reviewer that the range of isotopic values, instead of a specific value of a given $NH_3$ source should be served as input into isotopic mixing model. Unfortunately, the isotopic mixing model-IsoSource-used in the current study only allows a fixed isotopic value input for each $NH_3$ source. The good news is that the isotopic signatures of each source in our study have very narrow variation range, and there is only a few data overlap between the source signatures of waste and livestock (see figure above). These could significantly reduce the uncertainty of source apportionment by IsoSource. Besides, the uncertainty of model simulation also reported in Figure 3 in the MS. Nevertheless, we think future studies should include the adoption of more sophisticated Bayesian mixing models.

*Vehicle sampling Unlike the US, a major urban $NH_3$ source in China is human waste which has been found to have a $\delta^{15}N$-$NH_3$ value of ~ -41 to -30 per mil in this study. The vehicle exhaust in this study was sampled in a tunnel not directly from tailpipe. Most highly trafficked tunnels have ventilation systems that flush the tunnel with ambient air constantly. If the urban ambient air $NH_3$ is mainly from waste and the tunnels are flushed with ambient air, this mixing would lower the $\delta^{15}N$ value of the $NH_3$ sampled at the tunnel. Do these tunnels have ventilation systems? Could mixing with ambient air confound the $\delta^{15}N$ vehicle signal?*

The tunnel we chose does have ventilation system and the air in the tunnel is absolutely the mixture of vehicle exhaust and ambient air (even without the ventilation system). We think the mixing with ambient air (including $NH_3$ emissions from human excreta)

in the tunnel is not an important problem. Two reasons are as follows:

(1) The $NH_3$ conc. in the Tunnel (T-d in figure below; samples for isotopic analysis were also collected at) is nearly **11** times than that in ambient air ($O_{310m}$ and $O_{150m}$ in figure below) (Chang *et al.*, 2016). Therefore, it is safe to conclude that the $NH_3$ emissions in the Tunnel are dominated by vehicles instead of ambient air.

[Figure]

(**a**) Location of the eight sampling points in (labeled in yellow; inside the tunnel from the entrance to the exit) and out (labeled in green; varying in distance from the tunnel) of the Handan tunnel. The campus of Fudan University was separated into north and south parts by the tunnel. (**b**) Box-whisker plots of the $NH_3$ concentration sampled at each site, setting 20 as the breaking point of y axis. The box boundaries represent the 25th and 75th percentile, the horizontal line is the median, and the whiskers mark the 10th and 90th percentiles. (**c**) Relationship between the $NH_3$concentration at T-d (the exit of the Handan tunnel) and the other four sites varying in distance from the Handan road in the open environment.

(2) For $NH_3$ emission from human excreta, we recently quantified that that the population of ~21 million people living in the urban areas of Shanghai annually emitted approximately 1386 t $NH_3$, which corresponds to over 11.4% of the total $NH_3$ emissions in the Shanghai urban areas (Chang *et al.*, 2015). Therefore, we don't think human excreta is a major (but still important) urban $NH_3$ source in Chinese megacities like Beijing.

*Line 119: All three filters from a sampling event were combined for single analysis. Why?*

*Was there not enough N for analysis? This doesn't allow for reporting of the deviation, if any among samplers. Did the authors evaluate deviation among samplers? If so this should be included.*

Yes, ensuring enough N for isotopic analysis was our top priority. Efficient $NH_3$ PSD for short-term sampling and for N isotope analysis is still missing. In our previous work, two Ogawa filter samples collected monthly in Shanghai often cannot absorb enough $NH_3$-N for isotopic analysis. Although the ALPHA PSDs used in Beijing had larger filters to absorb more $NH_3$, the sampling time in our study was shortened. Therefore, we combined three ALPHA filters for a single analysis. Although we didn't have isotopic deviation among samplers, we had noticed the potential deviation of $NH_3$ concentrations among different filters. We had two co-located ALPHA PSDs simultaneously operating at our sampling site as part of our monitoring campaign. We didn't find any significant difference between the combined samples (for isotopic analysis) and these single filter samples.

*Line 166: When sampling exhaust from septic tanks the authors state "However, the $\delta^{15}N$-$NH_3$ values of daily samples varied widely (±10 per mil), suggesting that the isotope fractionation may occur during the process of sampling/storage. After many tests by trial and error, we found that a sampling period of 2 hours could provide sufficient N-$NH_3$ as well as avoid potential fractionation" It seems the daily samples would be more representative of the source and the fractionation that is occurring is representative of the source. The magnitude of fractionation is going to change under varying environmental conditions but this is a symptom of the source type and should be evaluated as the range in source signature.*

Compared with previous work, we have every confidence that we had optimized our sampling to provide sufficient N-$NH_3$ and avoid potential fractionation. As to the sampling period we chosen, please note that the $NH_3$ conc. in the ceiling ducts were very high. A whole day sampling was unpractical because the sampling filters could be overloaded. Besides, $NH_3$ conc. in the ceiling ducts only came from septic tanks without the interference of ambient air.

We cannot agree more that "the magnitude of fractionation is going to change under varying environmental conditions". If the fluctuation of daily environmental conditions could alter the $\delta^{15}N$-$NH_3$ values, then we can expect a much larger difference between different seasons. However, our results show that even sampling in different seasons, the $\delta^{15}N$-$NH_3$ values of the samples collected from septic tanks didn't show significant difference.

*Line 400: The authors state "However, as a direct product of NO reduction on the catalyst surface of TWCs ($2NO+5H_2 \rightarrow 2NH_3+2H_2O$ and/or $2NO+2CO+3H_2 \rightarrow 2NH_3+2CO_2$), $NH_3$ emitted from light-duty vehicle exhausts can be expected to have similar $\delta^{15}N$-values to vehicle-emitted $NO_x$." This reasoning is not*

*sound. NO produced may have a different original $\delta^{15}N$ value then the $NH_3$ being used in the TWC and the fractionation factor of the two different compounds caused by the TWC process could be very different. There is not valid evidence to state that the $\delta^{15}N$ of vehicle $NO_x$ and $NH_3$ would be the same.*

To avoid misunderstanding, we decided to delete this sentence in our revised MS (line 390-405).

Minor comments
*Line 28: APEC should not be abbreviated Line 51: delete "extensive" Line 160: "don't" should be replaced with "doesn't" Line 198: "A" should be deleted Lin329: The wording "far ahead" is not the appropriate descriptor here and should be changed.*

Revised accordingly. "far ahead" has been replaced by "much more than" in the revised MS.

*Figure 3: The x-axis labeling isn't sufficient. Are the boxes in order of sampling period? If so, provide the timeframe on the axis.*

Timeframe has been added on the axis.

---

## Author Comment (AC3) · 22 Aug 2016

The comment was uploaded in the form of a supplement:
http://www.atmos-chem-phys-discuss.net/acp-2016-432/acp-2016-432-AC3-
supplement.pdf
* * *

---

## Author Comment (AC4) · 22 Aug 2016

[revised manuscript text omitted]

---

## Author Response (AR1)

We thank two referees for their careful considerations of the manuscript and their well thought-out comments. These certainly helped to significantly improve the paper. We've addressed all comments and questions below in the form of point-by-point responses. The referees' comments are in *Italic* and our responses are in normal font. Text changes in the revised manuscript are highlighted in color (including a mistake we mentioned in our short comments).

**Referee 1:**

The manuscript by Chang et al. studied the sources of atmospheric ammonia before, during, and after the 2014 APEC summit in Beijing using stable nitrogen isotope signatures coupled with an isotope mixing model. The source contributions of traffic, waste, livestock, and fertilizer were quantified and compared. The results showed a substantial decrease (58.7%) of traffic emission during the APEC period with strict emission controls. The results also showed that non-agricultural sources (traffic and waste) of NH3 play an important role in particle pollution in the megacity of Beijing, which has important implications for future air pollution mitigating strategies. This is a pioneer study by applying isotopic measurements into source apportionment of NH3. Such an approach could be very powerful in future source apportionment studies particularly if combining with more collocated measurements. I recommend it for publication after addressing the following two comments.

Thanks for the encouraging comments, and also useful suggestions, which will certainly be taken into account in our field measurements in the future.

One of the assumptions of this study is that the contribution of biomass burning is considered minimal. The authors need to address the uncertainties of this assumption because Xu et al. (2015) showed that biomass burning contributed 12-19% to total organic aerosols before and during APEC in Beijing.

Firstly, biomass burning is well-known as a major source of ambient organic aerosols (OA). The factors/sources of OA in non-refractory submicron aerosols identified by aerosol mass spectrometer typically (including Xu *et al.* (2015)) include HOA, COA, biomass-burning (BBOA), coal combustion (CCOA), semi-volatile OOA (SV-OOA), and low volatility OOA (LV-OOA). Although biomass burning (e.g., crop residues, wild fires) also contribute to NH3 emissions, their emission factors are much less than that of OA and its large group of precursors (Akagi *et al.*, 2011; Stockwell *et al.*, 2014).

Secondly, we illustrated the locations (red dots) and number of fire spots (https://firms.modaps.eosdis.nasa.gov/) in Beijing and its neighboring region from 18th October to 29th November 2014. The figure below clearly showed that fire spots in each period were sparse, indicating the magnitude of open burning activities was very limited. Such a result can be expected because all crops grown in open-field (e.g. maize and cotton) in Northern China had been harvested before the APEC summit.

Thirdly, there is a considerable amounts of wood burning for heating in EU and US (Clark *et al.*, 2013; Saffari*et al.*, 2013). While this is not the case in Northern China where coal-based heating is overwhelmingly popular (Chen *et al.*, 2013). Besides, wood burning for cooking only exists in some rural areas of China that are far away from our sampling site in Beijing. Moreover, regions having wood burning for cooking also having intensive agricultural activities (e.g., livestock production), which makes NH3 emissions from wood burning relatively insignificant. In addition, the atmospheric behavior of NH3 is characterized by a short lifetime (1-5 days or less (Warneck, 1999)), low transport height, and relatively high dry deposition velocity (Asman and van Jaarsveld, 1992), high rural NH3 emissions do not generally influence urban areas strongly in the gaseous phase unless reacting with acidic gases locally to form particulate NH4+ (Flechard et al., 2013).

In conclusion, we are confident that the contribution of biomass burning to ambient NH3 concentrations was minimal during our sampling period in Beijing.

Reference:

- Akagi, S. K., Yokelson, R. J., Wiedinmyer, C., Alvarado, M. J., Reid, J. S., Karl, T., Crounse, J. D., and Wennberg, P. O.: Emission factors for open and domestic biomass burning for use in atmospheric models, Atmos. Chem. Phys., 11, 4039-4072, doi:10.5194/acp-11-4039-2011, 2011.
- Asman, W. A., and van Jaarsveld, H. A.: A variable-resolution transport model applied for NHx in Europe, Atmos. Environ., 26, 445-464, doi: 10.1016/0960-1686(92)90329-J, 1992.
- Chen, Y., Ebenstein, A., Greenstone, M., and Li, H.: Evidence on the impact of sustained exposure to air pollution on life expectancy from China's Huai River policy, Proc. Natl. Acad. Sci., 6, 110, 12936-12941, doi: 10.1073/pnas.1300018110, 2013.
- Clark, M. L., Peel, J. L, Balakrishnan K, Breysse, P. N., Chillrud, S. N., Naeher, L. P., Rodes, C. E., Vette, A. F., and Balbus, J. M.: Health and household air pollution from solid fuel use: the need for improved exposure assessment, Environ. Health Perspect., 121, 1120-1128, doi: org/10.1289/ehp.1206429, 2013.

- Flechard, C., Massad, R. S., Loubet, B., Personne, E., Simpson, D., Bash, J., Cooter, E., Nemitz, E., and Sutton, M. A.: Advances in understanding, models and parameterizations of biosphere-atmosphere ammonia exchange, Biogeosci., 10, 5183-5225, doi: 10.5194/bg-10-5183-2013, 2013.
- Saffari, A, Daher, N, Samara, C, et al.: Increased biomass burning due to the economic crisis in Greece and its adverse impact on wintertime air quality in Thessaloniki, Environ. Sci. Technol., 47, 13313-13320, doi: 10.1021/es403847h, 2013.
- Stockwell, C. E., Yokelson, R. J., Kreidenweis, S. M., Robinson, A. L., DeMott, P. J., Sullivan, R. C., Reardon, J., Ryan, K. C., Griffith, D. W. T., and Stevens, L.: Trace gas emissions from combustion of peat, crop residue, domestic biofuels, grasses, and other fuels: configuration and Fourier transform infrared (FTIR) component of the fourth Fire Lab at Missoula Experiment (FLAME-4), Atmos. Chem. Phys., 14, 9727-9754, doi:10.5194/acp-14-9727-2014, 2014.
- Warneck, P.: Chemistry of the Natural Atmosphere, pp. 426-441, Academic Press, San Diego, Calif., 1999.
- Xu, W. Q., Sun, Y. L., Chen, C., Du, W., Han, T. T., Wang, Q. Q., Fu, P. Q., Wang, Z. F., Zhao, X. J., Zhou, L. B., Ji, D. S., Wang, P. C., and Worsnop, D. R.: Aerosol composition, oxidation properties, and sources in Beijing: results from the 2014 Asia-Pacific Economic Cooperation summit study, Atmos. Chem. Phys., 15, 13681-13698, doi:10.5194/acp-15-13681-2015, 2015.

Coal combustion is the dominant source of aerosol particles during the heating season in Beijing. Unfortunately, such a source is missed in this study, which will affect the source apportionment results.

We appreciate the constructive criticism. We fully agree with the reviewer that coal combustion is the dominant source of aerosol particles during heating season in Beijing. However, we also think coal combustion-derived NH3 emissions could be largely neglected **in this study**.

First of all, the emission control measures taken by the Chinese government were really comprehensive and aggressive. For example, Beijing took half of the cars off local roads, closed more than 1000 heavy industrial plants within a 120-mile radius of the city. More importantly, although the convention was held during a heating period, coal-fired power plants operation and urban coal-based heating services in Beijing and Tianjin were suspended until the summit was closed (Zhang, 2014).

Besides, coal combustion is not necessary an important source of gaseous NH3. In fact, we have collected NH3 emissions from coal combustion in a combustion chamber by a glass-fritted bubbler system, and the initial results do not support the view that coal combustion is an important source of NH3 emissions (unpublished). Li *et al.* (2016) recently provided more robust evidence, suggesting that the average NH3 emission factors for burning 13 kinds of coal in a traditional heating stove was 1.01 mg g-1, and the advanced heating stove with a highly modified combustion efficiency had a much

lower average  $NH_3$  EF of 0.13 mg g-1. Supposing an amount of 19 Mt coal consumption in 2014, the annual  $NH_3$  emissions from coal combustion in Beijing was only 247-1919 t, which cannot be comparable with any other major  $NH_3$  sources (e.g., N fertilizer and animal manure/urine emissions).

Lastly, some may argue that coal-fired power plant (CFPP) NH3 slip is a major NH3 source in Beijing. But we do not think this can be true. In September 2013, a five-year plan was introduced in Beijing to slash coal consumption, and there were only four CFPPs operating near the city's urban areas during wintertime (China Daily, 2015). In 2016, all CFPP in Beijing will be shut down and replaced with gas-fired power plants to cut pollution. The replacement by the four gas-fired power plants will help cut emissions by 10000 t of sulfur dioxide and 19000 t of nitric oxide annually (China Daily, 2015). Although NH3 slip is a common issue with SCR (Selective Catalytic Reduction) technology used in CFPP for removal of nitric oxide, the mass concentration of ammonia (typically 3-5 mg NH3 m-3) in flue gases is two or three orders of magnitude smaller than that of NOx (MOE of China, 2014). Moreover, although there are many CFPPs surrounded Beijing in the North China Plain, most of which are co-located with intensive agricultural production areas.

**Reference:**

- China Daily: Beijing closes coal power plant to cut pollution, http://www.chinadaily.com.cn/china/2015-03/20/content\_19869722.htm. (accessible August 13, 2016)
- Li, Q., Jiang, J., Cai, S., Zhou, W., Wang, S., Duan, L., and Hao, J.: Gaseous ammonia emissions from coal and biomass combustion in household stoves with different combustion efficiencies, Environ. Sci. Technol. Lett., 3, 98-103, doi: 10.1021/acs.estlett.6b00013, 2016.
- MOE of China, Emission standard of air pollutants for thermal power plants (GB13223-2003), 2014.
- Zhang, S. R.: Beijing smog: The day after 'APEC Blue', 2014, http://thediplomat.com/2014/11/beijing-smog-the-day-after-apec-blue/. (accessible August 13, 2016)

**Referee 2:**

The authors have presented data which increases the inventory of isotopic signatures of ammonia emission sources, an area lacking in data. The authors have characterized isotopic signatures of NH3 emissions sources in China, a region of the world where this has not been done, and the data agrees closely with data obtained in other areas of the world. This data can be employed by researchers to quantify NH3 emissions contributing to ambient atmosphere. This is valuable as many NH3 emission sources are nonpoint sources making them difficult to quantify. Using the nitrogen isotope signatures the authors have estimated urban source contribution before and after a major event in Beijing in which air quality measures were employed during the event. The authors provide isotopic evidence that vehicles are a major contributor to urban *NH*3 concentrations and the isotopic signature in the city changed to reflect the reduction in vehicle emissions resulting from air quality measures. The estimation of other source contributions is likely confounded by potentially overlapping source signatures and this should be more adequately addressed by the authors. This a novel approach to assessing *NH*3 source contributions in an urban setting and the authors have presented the data in a clear and concise manner. If the major and minor issues below are addressed, I believe the manuscript could be accepted to *ACP*.

Thanks for the recognition of our contribution. Please check our point-by-point responses to the major and minor issues raised.

The authors use an isotope mixing model to predict NH3 source apportionment. The endmember signatures used in the model are vehicles, fertilizer, livestock waste, and human waste. The authors provide evidence that the vehicle endmember signature is significantly different from the other three endmembers but the other endmembers signatures are similar and with more sampling it is likely that these signatures will overlap. This is because the three sources are essentially the same "volatilized NH3" source and if the literature data is taken into consideration, these sources' signatures are observed to overlap. I don't think the authors have a viable case for assigning the source signatures they present to each source. I do however believe they could combine the "volatilized NH3" sources as one endmember and vehicles as another endmember. While this would only provide insight to vehicle source apportionment, I think it is a more realistic approach. If the authors keep the mixing model as is, they need to explain the caveats associated with the estimates of the "volatilized NH3" sources and include error analysis. Rather than reporting specific values a range should be reported that represents the deviation and error involved in the calculations.

(1) We agree with the reviewer's comments that some of the isotopic signatures of volatilized sources could overlap and this problem should be addressed properly.

We think the source classification in our study is reasonable.

Our NH3 source classification (Figure below) is based on:

a. The difference of isotopic signatures.

In our MS we concluded that "NH3 emitted from volatilized sources has relatively low  $\delta^{15}$ N values, allowing them to be distinctly differentiated from NH3 emitted from traffic sources that are characterized by relatively high  $\delta^{15}$ N value". Obviously, traffic-derived and fertilizer-volatized NH3 has the highest and the lowest  $\delta^{15}$ N values, respectively. However, "three sources are essentially the same volatilized NH3" does not necessarily signify that all volatilized NH3 has a similar variation range of  $\delta^{15}$ N-NH3 values (data overlap), nor does it mean that these volatilized NH3 sources should be classified as a single source category. Temperature is an important parameter, but not the only parameter in determining the differences of  $\delta^{15}$ N-NH3 values among different NH3

sources. In other words, even if data overlap may occur within volatilized sources, they can still be further classified as several sub-categories because of their fundamentally different emission mechanisms (e.g., organic N *vs.* inorganic N; see discussion later);

Variation range of isotopic signatures for different NH3 sources in our study.

b. The results of emission inventories or the practical emission situation of our targeted study area.

If our sampling site is located in rural areas, then we agree with the reviewer that all volatilized sources can be combined as a single source because NH3 source contributions in rural areas are much simpler: agricultural volatilized sources predominate the NH3 emission budget. However, in urban areas, except for on-road traffic, there are many other important non-agricultural NH3 sources (also volatized sources). Urban wastewater treatment plants and solid waste are arguably two of the most important NH3 sources in urban China. As a case study, we've identified that human excreta stored in septic tanks in Shanghai is a stable and important source of atmospheric NH3, contributing to over 11% of the total NH3 emissions in the Shanghai urban areas. Therefore, it is of critical importance to incorporate the isotopic signatures of waste-related NH3 sources into the isotope mixing model.

More importantly, even in natural/rural areas, we still need to take account of emission inventory information. If we combined all volatilized sources as a single source, the underlying logic is that every volatilized source contributes equally to ambient NH3 concentrations. However, wastewater and solid waste, generally, have an insignificant contribution to the rural NH3 budget. Besides, if we simply separate all NH3 sources into two categories (volatilized source and combustion source) without the consideration of the emission inventory, then we are highly likely to get into a trouble: some  $\delta^{15}$ N values of ambient NH3 sources.

To facilitate our explanation, we would like to give a counterexample. In a recent paper entitled "Fossil fuel combustion-related emissions dominate atmospheric ammonia sources during severe haze episodes: Evidence from 15N-stable isotope in size-resolved aerosol ammonium", Pan *et al.* (2016) presented the isotopic measurements of size-resolved aerosols in Beijing, summarizing that fossil fuel-related ammonia emissions (including traffic, coal combustion and power plants NH3 slip) have overtaken

agricultural activities as the dominant source of atmospheric NH3 during the hazy days of 2013.

There are three NH3 sources, i.e., agricultural NH3 volatilization, fossil fuel combustion, and power plant NH3 slip, considered in Pan *et al.* (2016), and the average  $\delta^{15}$ N values (specific values) of these sources were used as isotope signatures ( $\delta^{15}$ N-NH3) to estimate their relative contributions to the ambient NH3 in Beijing. We fully understand the author's consideration in terms of the classification of NH3 sources: agricultural ammonia is emitted at environmental temperature, the process of fossil fuel combustion can directly emit ammonia at high temperature, and ammonia slipped from power plant is the residue of gaseous reductants (typically anhydrous ammonia, aqueous ammonia or urea) that are subjected to medium temperature, and the isotopic signatures of these sources can be separated based on various temperatures.

However, we have to point out that temperature-only-based NH3 source classification has a few severe problems, which may lead to wrong conclusions and therefore could potentially mislead China's policy on future NH3 emissions reduction.

It is well accepted that agricultural activities-fertilizer application merged with livestock production-are the largest contributors of NH3 emissions at a regional or global scale. However, recent works reveal that NH3 volatilized from fertilizer application and livestock waste have distinct  $\delta^{15}$ N values, which can also be reflected by the large variation range of  $\delta^{15}$ N values for agricultural source in Pan *et al.* (2016) (see Figure 3). This is because fertilizer application and livestock waste generally represent two totally different nitrogen forms, i.e., inorganic and organic nitrogen, respectively. Situated on the northern edge of the North China Plain, one of the most intensive agricultural regions in China, Beijing is regarded as a receptor of agricultural ammonia from rural areas. In Pan *et al.* (2016), fertilizer application and livestock waste were combined as a single source (volatilized source), which could inevitably underestimate the contribution of agricultural activities to the ambient NH3 in Beijing.

If temperature is a decisive factor, then power plant NH3 slip considered as a major NH3 source in Pan *et al.* (2016) should be reasonable. However, Pan *et al.* (2016) claimed that during haze periods, **49%** of NH3 in the ambient atmosphere of Beijing was derived from power plant NH3 slip, which can hardly be true. In September 2013, a five-year plan was introduced in Beijing to slash coal consumption, and there were only four coal-fired power plants (CFPP) operating near the city's urban areas during wintertime. In 2016, all CFPP in Beijing will be shuttled and replaced with gas-fired power plants to cut pollution. The replacement by the four gas-fired power plants will help cut emissions by 10000 t of sulfur dioxide and 19000 t of nitric oxide annually. Although ammonia slip is a common issue with SCR (Selective Catalytic Reduction) technology used in CFPP for removal of nitric oxide, the mass concentration of ammonia (typically 3-5 mg NH3 m-3) in flue gases is two or three orders of magnitude smaller than that of NOx. Moreover, it is necessary to consider that although there are many CFPPs

surrounding Beijing in the North China Plain, most of these are co-located with intensive agricultural production areas.

(2) We also fully agree with the reviewer that the range of isotopic values, instead of a specific value of a given NH3 source should be served as input into isotopic mixing model. Unfortunately, the isotopic mixing model-IsoSource-used in the current study only allows a fixed isotopic value input for each NH3 source. The good news is that the isotopic signatures of each source in our study have very narrow variation range, and there is only a few data overlap between the source signatures of waste and livestock (see figure above). These could significantly reduce the uncertainty of source apportionment by IsoSource. Besides, the uncertainty of model simulation also reported in Figure 3 in the MS. Nevertheless, we think future studies should include the adoption of more sophisticated Bayesian mixing models.

Vehicle sampling Unlike the US, a major urban NH3 source in China is human waste which has been found to have a  $\delta^{15}$ N-NH3 value of ~ -41 to -30 per mil in this study. The vehicle exhaust in this study was sampled in a tunnel not directly from tailpipe. Most highly trafficked tunnels have ventilation systems that flush the tunnel with ambient air constantly. If the urban ambient air NH3 is mainly from waste and the tunnels are flushed with ambient air, this mixing would lower the  $\delta^{15}$ N value of the NH3 sampled at the tunnel. Do these tunnels have ventilation systems? Could mixing with ambient air confound the  $\delta^{15}$ N vehicle signal?

The tunnel we chose does have ventilation system and the air in the tunnel is absolutely the mixture of vehicle exhaust and ambient air (even without the ventilation system). We think the mixing with ambient air (including NH3 emissions from human excreta) in the tunnel is not an important problem. Two reasons are as follows:

(1) The NH3 conc. in the Tunnel (T-d in figure below; samples for isotopic analysis were also collected at) is nearly 11 times than that in ambient air (O310m and O150m in figure below) (Chang *et al.*, 2016). Therefore, it is safe to conclude that the NH3 emissions in the Tunnel are dominated by vehicles instead of ambient air.